# WINDS Model Demonstration with Field Data from a Furrow-Irrigated Cotton Experiment

Hadiqa Maqsood [1], Douglas J. Hunsaker [2], Peter Waller [1,*], Kelly R. Thorp [2], Andrew French [2], Diaa Eldin Elshikha [1] and Reid Loeffler [1]

[1] Department of Biosystems Engineering, University of Arizona, Tucson, AZ 85721, USA; hadiqa@arizona.edu (H.M.); diaaelshikha@arizona.edu (D.E.E.); reidloeffler@arizona.edu (R.L.)

[2] USDA-ARS, Arid Land Agricultural Research Center, Maricopa, AZ 85138, USA; doug.hunsaker@usda.gov (D.J.H.); kelly.thorp@usda.gov (K.R.T.); andrew.french@usda.gov (A.F.)

* Correspondence: pwaller@arizona.edu; Tel.: +1-520-440-5803

**Abstract:** The WINDS (Water-Use, Irrigation, Nitrogen, Drainage, and Salinity) model was developed to provide decision support for irrigated-crop management in the U.S. Southwest. The model uses a daily time-step soil water balance (SWB) to simulate the dynamics of water content in the soil profile and evapotranspiration. The model employs a tipping bucket approach during infiltration events and Richards' equation between infiltration events. This research demonstrates WINDS simulation of a furrow-irrigated cotton experiment, conducted in 2007 in central Arizona, U.S. Calibration procedures for WINDS include the crop coefficient curve or segmented crop coefficient curve, rate of root growth, and root activity during the growing season. In this research, field capacity and wilting point were measured in the laboratory at each location and in each layer. Field measurements included water contents in layers by neutron moisture meter (NMM), irrigation, crop growth, final yield, and actual $ET_c$ derived by SWB. The calibrated WINDS model was compared to the neutron probe moisture contents. The average coefficient of determination was 0.92, and average root mean squared error (RMSE) was 0.027 m³ m⁻³. The study also demonstrated WINDS ability to reproduce measured crop evapotranspiration ($ET_c$ actual) during the growing season. This paper introduces the online WINDS model.

**Keywords:** irrigation scheduling; model; evapotranspiration; soil water; root activity; cotton

## 1. Introduction

In the southwestern United States (U.S.), agroecosystems are experiencing water stress due to overallocation of the agricultural water supply, ongoing drought in the Colorado River basin, depletion of groundwater storage and reservoir water supplies, and increased distribution of water for industrial and municipal needs [1–4]. Efficient irrigation management is needed to optimize crop yields and minimize the amount of water application and water loss. Irrigation field experiments conducted in the region for the past decades have developed strategies for improving crop yield and water management [5–9]; however, with the growing strain on water supplies, there is renewed urgency to develop improved decision support models for irrigated systems in the southwestern U.S.

Soil water balance (SWB) models can evaluate experiments and provide decision support for planning, irrigation scheduling, and crop management [10]. Models can evaluate deficit irrigation strategies in order to optimize water use efficiency [11–13]. Several studies have compared various soil water balance/crop models [14,15]. Two Food and Agriculture Organization (FAO) daily time-step models, CropWat [16] and AquaCrop [17], are popular irrigation and crop growth models. AquaCrop is a daily time-step soil water balance model. It calculates crop growth and yield as a function of water stress. Use of AquaCrop has been reported to improve irrigation scheduling and improve water use efficiency [18].

Finite difference models such as SWMS [19] and HYDRUS 1D, 2D, and 3D [20–22] accurately simulate soil moisture; however, these data-intensive models are not suitable for daily irrigation management.

The WINDS (Water-Use, Irrigation, Nitrogen, Drainage, and Salinity) model was developed as a decision support model for crop growers in the U.S. Southwest. A theoretical description of the WINDS model is in the textbook *Irrigation and Drainage Engineering* [23]. The irrigation model in WINDS uses a daily time-step SWB and integrates a tipping bucket approach to simulate water distribution in sequential layers of the soil profile layers after irrigation and rainfall events, whereas the water movement between infiltration events is simulated with Richards' equation [24]. The WINDS model calculates crop evapotranspiration (ET$_c$) with the FAO56 dual crop coefficient procedures [25]. Figure 1 is a schematic illustration of the soil water processes in the WINDS irrigation model that shows soil-layer-based wetting fractions and distribution of available water and overall SWB in the profile. The WINDS model parameterizes changes in the root activity of sequential soil layers during the growing season. Thus, the strength of the WINDS model is that it can simulate soil water content changes in discrete layers of the soil profile. This increases the usefulness of the model for irrigation management because the model can also be periodically updated with data from in situ soil water sensors at specific depths in the profile. Although it is impossible to accurately characterize spatially varying soil properties in a field, the WINDS model can be parameterized for general soil types and for the layer in which there are sensor measurements. Although there are excellent daily soil water balance models such as Aquacrop in use around the world, the WINDS model focuses on the irrigation systems, evapotranspiration, soil water status in layers, and crops in the arid southwestern United States. Further description of the WINDS irrigation model is presented in Section 2.3.

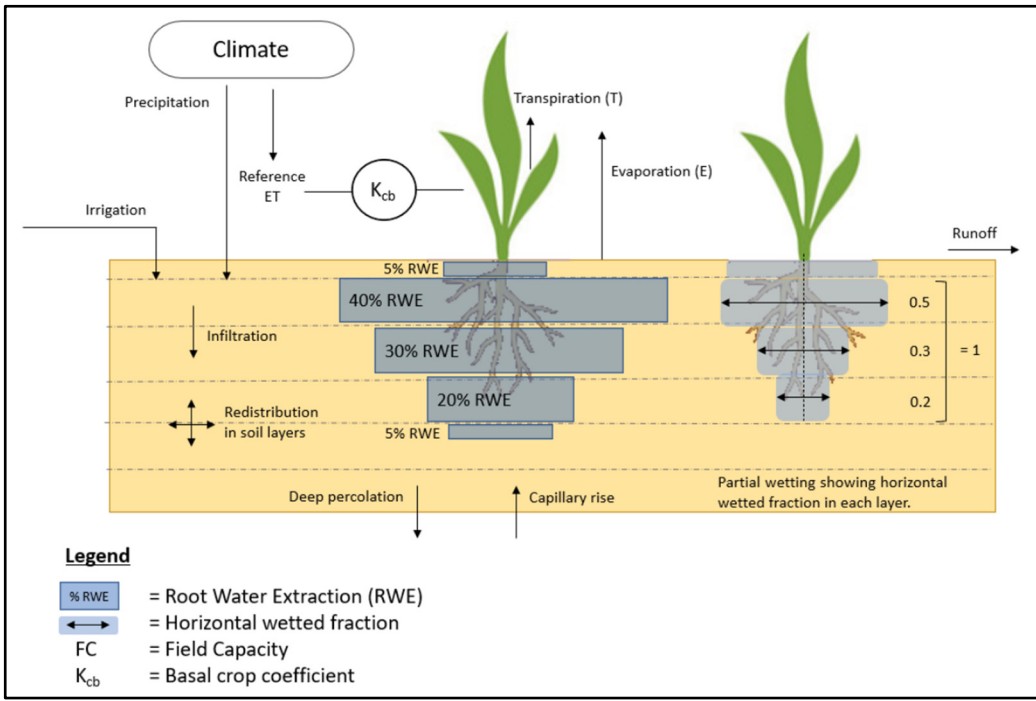

**Figure 1.** Schematic diagram of the WINDS irrigation model.

Ground-based and aerial remote sensing (RS) are effective tools for crop management and characterization. RS data can provide averages for fields, as well as spatially varying data within fields [26]. Applications of RS are considerable, e.g., from characterizing plant growth and crop coefficients to aiding in irrigation scheduling decisions for improved water productivity [27–30]. Currently, sensors (cameras) with bands in the visible, near-

infrared, and thermal parts of the electromagnetic spectrum are commonly installed on tractors, drones, and satellites. The normalized difference vegetation index, NDVI [31], provides a measure of crop biophysical parameters and growth [7]. Many studies have used indices such as NDVI to estimate the basal crop coefficient ($K_{cb}$) within the FAO56 $ET_c$ procedures [32–34]. With cotton, Hunsaker [27] compared the accuracies of an NDVI-based $K_{cb}$ approach and the FAO56 segmented $K_{cb}$ curve approach. That study showed good agreement between the measured $ET_c$ actual and estimated $ET_c$ by both $K_{cb}$ methods.

Cotton ranks seventh in cultivated area in the world and is in high demand in textile and chemical industries. Cotton has been grown commercially in the southwestern U.S. desert of Arizona for more than 100 years, most commonly using surface irrigation systems, such as furrow and level basin. Because it has high water demand in the desert, various researchers over the years have studied irrigation water use and efficiency in cotton [35–37]. Experiments on cotton in the low desert region have shown that higher yields and water savings can be realized with smaller but more frequent surface irrigation applications [38,39]. More recently, various crop modeling techniques are being evaluated in cotton research experiments to simulate management with decreasing water availability and climate uncertainty [40].

The primary objective of this study was to evaluate the performance of the WINDS model for simulation of daily soil water content in soil layers with soil parameters measured in the laboratory and with calibrated root growth and water extraction patterns. Data were from a surface irrigated cotton field experiment in central Arizona. The experiment included detailed measurements of soil water content, irrigation, soil, plant, and remote sensing parameters.

## 2. Materials and Methods

### 2.1. Cotton Field Experiment Layout

The cotton irrigation experiment was conducted in 2007 on a 1.25 ha field site at the University of Arizona, Maricopa Agricultural Center (MAC) in central Arizona (33°04′ N, 111°58′ W, 361 m MSL). The climate is hot and dry, except for the summer monsoon season in July through September. The soil was classified (Soil Taxonomy, 2006) as Casa Grande series (Typic Natrargids, fine-loamy, mixed, hyperthermic) by Post et al. [41] which consists of sandy loam to sandy clay loam soil textures. Before the cotton experiment, Camelina sativa was planted in early November 2006 and grown with irrigation in 32 treatment plots [42]. The camelina crop was harvested on 19 April 2007 (day of year (DOY) 109). The field was then ripped, disked, and laser-leveled to zero slope to prepare for cotton planting. On 27 April, ammonium phosphate (16-20-0) at rates of 36 and 45 kg N ha$^{-1}$ was incorporated into the soil as pre-plant fertilizer, followed by soil cultivation, and bed listing in the N-S direction at 1.02 m spacings. On 30 April (DOY 120), the cotton cultivar DeltaPine-449B/RR was planted in the dry raised beds of 32 plots using a four-row, precision vacuum planter (Monosem Inc., Edwardsville, KS, USA). Two seeding rates, typical (T), 12 seeds m$^{-2}$, and sparse (S), 6 seeds m$^{-2}$, were each applied in 16 of the 32 plots.

The experimental field contained eight sub-treatment plots that were randomly assigned in a randomized complete block design (RCBD) in four replicate blocks (32 total treatment plots). The main treatment included four irrigation scheduling methods, each having sub-treatments of the T and S plant densities. Irrigation scheduling methods included two treatments in which daily $ET_c$ was estimated using the FAO56 dual crop coefficient procedures based on two $K_{cb}$ approaches: (1) a segmented FAO56 (F) $K_{cb}$ curve based on local research and (2) daily $K_{cb}$ estimated from NDVI (N) observations that were made two to three times per week and interpolated daily. There were two other treatments, but for this study, only treatments using the two Kcb methods under typical plant density (denoted as FT and NT, respectively) were evaluated.

After planting, the plots were separated by forming raised berms on the four edges of each plot. After berming, the inner plot size of each replicate was 11 rows wide (11.2 m) by 18.5 m long. Figure 2 and Table 1 show the assigned field plot numbers of the four replicates

of each treatment. The other unmarked plots in Figure 2 were the different experimental treatments not evaluated herein. Two gated pipe irrigation systems were installed in the E-W direction and extended the length of the field. Irrigation water was controlled by an alfalfa valve located at the west end of each gated pipe system. Irrigation volume was measured in each plot using calibrated in-line propeller-type water meters placed at the head of each gated pipe system. Metal neutron access tubes were then installed vertically to a depth of 3.0 m in central beds in each plot at approximately the geometric center of the plot. Two steel rods, 0.3 m long, used in measuring surface depth volumetric soil water content ($\theta_v$, $m^3$ $m^{-3}$) by a time-domain reflectometry (TDR) sensor (Trase1 Soil Moisture Equip., Corp., Santa Barbara, CA, USA) were also installed vertically in the same plot bed as the access tube, but 0.5 m away from the tube. In-season $\theta_v$ measurements for plots were made with the TDR (0 to 0.3 m soil depth) and by field-calibrated neutron moisture meters (NMMs) (CPN 503, Instrotek, Co., Concord, CA, USA) from 0.3 to 2.9 m, in 0.20 m increments. Initial measurements were made on 1 June (DOY 152), one month after planting. Thereafter, 26 $\theta_v$ readings for the FT and NT plots were made every 5–7 days through 1 October 2007 (DOY 274). The $\theta_v$ at planting (DOY 120) for plot replicates were estimated from readings made in the same plots at the camelina harvest (DOY 109) by the same NMM instruments. No precipitation occurred between DOY 109 and 120. The soil properties for plots were obtained in the prior camelina experiment and were provided in [42].

**Table 1.** Treatments and plot numbers for cotton 2007 experiment.

| Treatment | Plot Number | Title 3 |
|---|---|---|
| FAO method | 12 | FT1 |
| | 25 | FT2 |
| | 34 | FT3 |
| | 46 | FT4 |
| NDVI method | 24 | NT1 |
| | 26 | NT2 |
| | 31 | NT3 |
| | 36 | NT4 |

Canopy spectral reflectance factors were collected bi-weekly over all plots (from DOY 152 to 276). Measurements were made using a 4-waveband Exotech radiometer (Model BX-100; Exotech, Inc., Gaithersburg, MD, USA) equipped with 15° field-of-view optics and held in a nadir orientation approximately 1.5 to 2.0 m above the soil surface. The NDVI was computed as

$$NDVI = (NIR - Red)/(NIR + Red) \tag{1}$$

where the range of NDVI is from 0 to 1, where 0 represents no live green vegetation, and 1 represents fully covered live green vegetation. Reflectance data obtained when there was cloud interference with the direct beam solar insolation or when soils were wet from irrigation or rainfall were not used.

All FT and NT plots achieved full cover by 23 August (DOY 235). All treatments were terminated with defoliants on DOY 280 making the growing season 160 days long. The crop was harvested between DOY 297 and 299. Final yields were obtained from seed cotton that was hand-picked within a final harvest area measuring approximately 24 $m^2$ in the south half of each plot and then ginned at the MAC ginning facility.

Daily weather data for 2007 were provided by The Arizona Meteorological Network (AZMET; https://cals.arizona.edu/AZMET/index.html) weather station at MAC, located about 200 m from the field site. The AZMET station also provided daily, short-crop reference evapotranspiration ($ET_o$) using the standardized FAO56 equation [25]. During the 2007 cotton season, June, July, and August were the warmest months (Table 2). The highest daily maximum temperature recorded was 45.7 °C on 4 July 2007. July and August received frequent rainfall events with July being the wettest month in the growing period with a

total of 45 mm of rainfall (Table 2). The drier months of May and June had a higher average $ET_o$ than in August when the summer monsoon season was in full force.

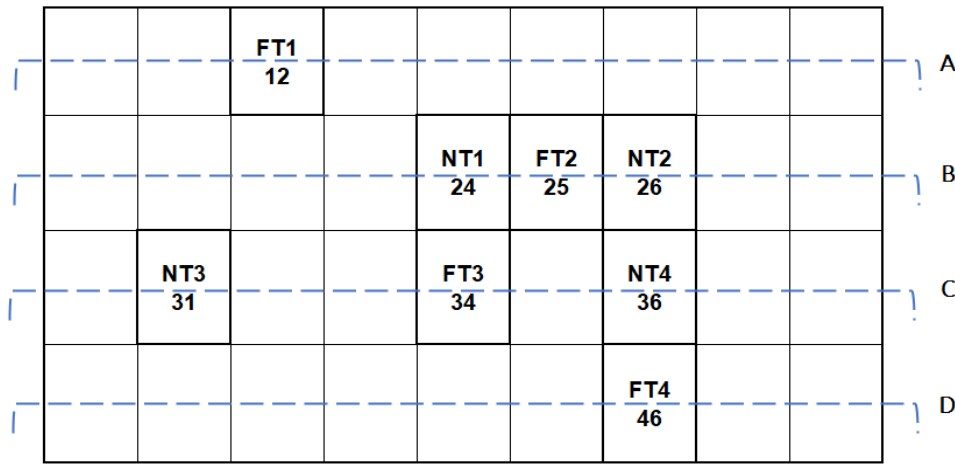

**Figure 2.** Layout of the field experiment showing the replicates for the FAO (FT) and NDVI (NT) replicates (1–4) under study. Plots are 11.2 m (horizontal directions) × 18.5 m (vertical direction). A, B, C, and D represent transects in Figure 3.

**Figure 3.** Soil types in layers for plots in study. The hatched lines represent mixed soils. B, C, and D represent transects shown in Figure 2.

**Table 2.** Daily average weather parameters by month and cumulative monthly rainfall and reference evapotranspiration ($ET_o$) at Maricopa, Arizona, for the growing period of cotton in 2007.

| Month | Max Temp (°C) | Min Temp (°C) | Rainfall (mm) | $ET_o$ (mm) | Wind Speed 2 m (m/s) |
|---|---|---|---|---|---|
| May | 35.5 | 16.9 | 0.0 | 245.5 | 2.1 |
| June | 40.4 | 21.0 | 0.0 | 265.3 | 2.1 |
| July | 40.4 | 25.4 | 45.0 | 244.4 | 2.2 |
| August | 40.1 | 25.6 | 9.7 | 210.1 | 1.7 |
| September | 37.5 | 20.4 | 2.3 | 185.9 | 1.9 |
| Total growing period ave. | 37.7 | 20.3 | 60.0 | 1151 | 2.0 |

The crop was seeded into dry soil. Initial irrigations were conducted in the days after planting (DOY 122–130), with a total of 327 mm applied. Subsequent irrigations were conducted on DOY 155 (55 mm), 169 (68 mm), 180 (77 mm), 190 (90 mm), 204 (119 mm), 221 (112 mm), 233 (113 mm), and 248 (77 mm). Slightly different amounts were applied to different plots, as affected by the calculated SWB irrigation scheduling procedures presented next.

*2.2. Field Experiment Irrigation Scheduling and Crop ET Evaluation*

Irrigation scheduling of the FT and NT plot replicates was based on a daily SWB model of the cotton rooting depth ($Z_r$), which estimated root zone depletion ($D_r$) at the end of each day (Equation (2)):

$$D_{r,i} = D_{r,i-1} - P_i - I_i + ET_{c,i} + DP_i \tag{2}$$

where $D_{r,i}$ and $D_{r,i-1}$ are the root zone depletion (mm) at the end of day i and day i − 1, respectively, and $P_i$, $I_i$, $ET_{c,i}$, and DPi are amounts of precipitation, net irrigation depth, crop ET, and deep percolation, respectively, on day i, all in units of mm. The values for $I_i$ were the measured irrigation depths given to the particular plot replicate. Daily values of total available water ($TAW_i$) of the daily rooting depth ($Z_{r,i}$) were calculated as

$$TAW_i = 1000\, Z_{r,i}\, (FC - WP) \tag{3}$$

where $TAW_i$ is in mm, $Z_{r,i}$ is in m, and FC and WP are the soil profile average field capacity and wilting point values, respectively, determined separately for each plot by Hunsaker [42] in $m^3\ m^{-3}$. The limits for $D_{r,i}$ in Equation (2) are zero (at FC) and TAW (at WP). Since $D_{r,i}$ cannot be less than zero on a given day i following irrigation and/or precipitation, an amount for $DP_i$ was computed, when necessary, to balance Equation (2), if $D_{r,i}$ was less than zero on day i. The $Z_{r,i}$ were increased from an initial value of 0.60 m at planting to a maximum of 1.8 m at mid-season. Increase of $Z_{r,i}$ was based on observed soil water extraction patterns. Initiation of the SWB was on the day of planting where the initial $D_{r,i}$ was near WP for plots, based on the $\theta_v$ data prior to planting (i.e., at camelina harvest). Irrigation commenced on the day after planting.

The FAO56 dual crop coefficient procedures [25] were employed to compute estimated daily $ET_c$ ($ET_{c,i}$) for both the FAO and NDVI treatments (Equation (4)):

$$ET_{c,i} = (K_{cb,i}\, K_{s,i} + K_{e,i})\, ET_{o,i} \tag{4}$$

where $K_{cb,i}$ and $K_{e,i}$ are the daily basal crop and evaporation coefficients, respectively, and $K_{s,i}$ is a stress coefficient that introduces the effects of water stress on $ET_{c,i}$, where $K_{s,i} < 1$ when the available soil water is insufficient for full $ET_c$, and $K_{s,i} = 1$ when there is no soil water limitation. Daily $K_{cb}$ was estimated for the FT treatment plots using a single, locally developed 4-segment $K_{cb}$ curve as derived by [27]. The parameters describing the $K_{cb}$ curve for FT are given in Table 3. For the NT treatment, daily $K_{cb}$ was calculated using prediction equations that express $K_{cb}$ as a function of NDVI. For each NT replicate, after

interpolating the NDVI observations over the plot on a daily basis (i.e., $NDVI_i$), a daily $K_{cb}$ was calculated. The $K_{cb}$-NDVI relationships used for the NT plots are shown in Table 3.

**Table 3.** Parameters describing the FAO56 basal crop coefficient ($K_{cb}$) curve and the NDVI-based $K_{cb}$ used in irrigation scheduling for the 2007 cotton experiment in Maricopa, Arizona.

| Growth Stage | Day of Year | Growth Stage Length (days) | Estimated $K_{cb}$ | | Estimated Crop ET (mm) | |
|---|---|---|---|---|---|---|
| | | | FAO56 | NDVI | FAO56 [a] | NDVI [a] |
| Initial stage | 122–151 | 30 | 0.15 | 0.16 | 95.8 ± 4.9 | 103.7 ± 7.3 |
| Development | 152–203 | 52 | 0.17–1.18 | 0.28–1.16 | 362.3 ± 3.1 | 359.3 ± 15.5 |
| Mid-season | 204–253 | 50 | 1.20 | 1.21 | 400.7 ± 2.7 | 399.4 ± 1.2 |
| Late season | 254–276 | 23 | 1.17–0.51 | 1.20–0.26 | 83.1 ± 15.8 | 91.3 ± 1.3 |
| Total | 122–276 | 155 | | | 941.8 ± 24.5 | 953.8 ± 12.1 |

| NDVI-based $K_{cb}$ criteria | |
|---|---|
| Growth stage | $K_{cb}$-NDVI relationship [b] |
| Initial through mid-season | $K_{cb} = -0.21 + 5.0 \times NDVI - 12.2 \times NDVI^2 + 14.9 \times NDVI^3 - 6.2 \times NDVI^4$ |
| Late season | $K_{cb} = -125 + 498 \times NDVI - 662 \times NDVI^2 + 294 \times NDVI^3$ |

Notes: [a] Average plus and minus one standard deviation for the four replicates in the treatment. [b] From Hunsaker [27].

The FT and NT treatments were managed to replace $ET_{c,i}$ with I and/or P, while maintaining the SWB such that the daily soil water depletion ($SWD_i$, Equation (5)) $\approx < 45\%$ (irrigation trigger):

$$SWD_i = 100 \times [1 - (TAW_i - D_{r,i})/TAW_i] \tag{5}$$

where $SWD_i$ is in percent, and $TAW_i$ and $D_{r,i}$ are previously defined. A daily water stress coefficient ($K_{s,i}$) was calculated by Equation (6):

$$K_{s,i} = (TAW_i - D_{r,i})/((1 - p)\,TAW_i) \tag{6}$$

where $K_{s,i}$, $TAW_i$, and $D_{r,i}$ are as previously defined; and p is the fraction of $TAW_i$ that can be extracted from the root zone without suffering water stress. The p value was not adjusted daily but was a baseline value of 0.65 for cotton [25].

Daily values of $K_e$ ($K_{e,i}$) were estimated following irrigation or precipitation by computing Equation (7) (FAO56, Equation (7)), which requires a daily SWB of the surface evaporative soil layer (assumed as 0.1 m in thickness).

$$K_{e,i} = \min\,[K_{r,i}\,(K_{cmax,i} - K_{cb,i}),\, f_{ew,i} \times K_{cmax,i}] \tag{7}$$

The $K_{r,i}$ are the daily evaporation reduction coefficients dependent on the daily cumulative depth of evaporation ($D_{e,i}$) from the surface layer on day i following complete wetting of the surface. Total evaporable water (TEW) of the surface layer (0 to 0.10 m) was calculated using the measured FC and WP values of the top 0.3 m soil depth for each plot and averaged 18.9 ± 2.2 and 21.2 ± 2.7 mm for the FT and NT replicates, respectively. The readily evaporable water (REW) was assumed as 9 mm for all plots, an estimated value for a sandy loam soil [25]. The $K_{cmax,i}$ are the daily maximum values of $K_{c,i}$ that can occur following $I_i$ or $P_i$. Calculation of $K_{cmax,i}$ requires values for daily minimum relative humidity and average wind speed, provided by AZMET, and crop height, which was increased from a minimum value of 0.05 m at the initial growth stage to a maximum of 1.2 m at mid-season, in proportion to the increase in $K_{cb,i}$ from minimum to maximum values. The $f_{ew,i}$ values are the daily fractions of soil, both wetted and exposed. The $f_{ew,i}$ were determined from estimates of daily covered soil fraction ($f_{c,i}$) and fraction of soil wetted ($f_{w,i}$) by $I_i$ or $P_i$. The $f_{c,i}$ were increased from a minimum value of 0.0 at the initial stage of growth to 0.98 at mid-season, in proportion to the increase in $K_{cb,i}$. The $f_{w,i}$ values were assumed as 0.50 for alternate furrow irrigation and as 1.0 for regular every-furrow irrigation, as well as for precipitation events.

Actual (measured) cotton ET was calculated as the residual of the root zone SWB (Equation (8)) for periods bounded by two adjacent dates of $\theta_v$ measurements:

$$ET_{c\ act} = (D_{r,2} - D_{r,1}) + I + P - DP \tag{8}$$

where $ET_c$ act is the total actual $ET_c$ (mm) that occurred in the period from the first (1) to second (2) measurement date, and $D_{r,1}$ and $D_{r,2}$ are the measured root zone soil water depletion (mm) on the first and second date, respectively. The I, P, and DP, respectively, are total depth of measured irrigation (mm), total measured precipitation (>1.5 mm per day), and total deep percolation (not measured) below the root zone (mm) that occurred during the period. The $\theta_v$ was measured to a 2.9 m depth to allow detection of any water percolating below the cotton root depth (maximum of 1.8 m). When irrigation or significant rainfall (>10 mm) occurred between two soil water measurement dates, an estimate was made for the amount of DP that may have occurred below the 1.8 m root zone following the water input. This estimate was calculated as the summation of the quantity of increased soil water storage that occurred from soil layers below 1.8 m:

$$DP = \sum_1^6 (\theta 2 - \theta 1)\,\Delta S \tag{9}$$

where DP is in mm, $\theta_1$ and $\theta_2$ are the volumetric soil water contents (m$^3$ m$^{-3}$) before and after the wetting event, respectively, for each of the six soil layers in the profile measured below 1.8 m, and $\Delta S_i$ is the thickness (0.20 m) of each soil layer. Positive values by Equation (9) were added as DP in Equation (8), whereas zero or negative results indicated negligible moisture movement below the root zone, and DP was set to zero for those periods.

*2.3. WINDS Model*

A description of the WINDS model is in *Irrigation and Drainage Engineering* [23]. This study used the Excel/VBA version of WINDS for the analysis but added the information to the new online version. In this research, partial wetting of the soil profile in the horizontal direction was added to the WINDS model because of the need to simulate alternate furrow irrigation in the early season of the cotton experiment.

The WINDS model has different soil water flux algorithms for irrigation events (tipping bucket) and the periods between (Richards' equation) irrigation events. During large infiltration events, the wetting front has large changes in water content and energy over short distances and time intervals. Simulation based on energy and water gradients with Richards' equation would require small time and space steps and excessive computation time; thus, the WINDS tipping bucket algorithm routes large infiltration events downward through the root zone based on available water capacity in each modeled soil layer without considering water and energy gradients. There are two tipping bucket algorithms in the WINDS model. In light soils, all layers drain to FC on the same day as the irrigation event. In heavy soils, the downward water movement in any layer cannot exceed the saturated hydraulic conductivity in that layer. In this research in light soils, there was no restriction by hydraulic conductivity on infiltration, and all soil layers drained to FC capacity on the irrigation day.

The upper soil layer is the evaporation layer. Equation (10) is the water balance for the evaporation layer with partial horizontal wetting, where sources are irrigation and rain infiltration, and sinks are evaporation and transpiration. In the case of partial horizontal with alternate furrow irrigation, the WINDS model routes water between partially wetted layers with varying widths (Figure 1). In the evaporation layer, evaporation ($E_{wet}$) and rain infiltration ($R_{wet}$) are adjusted based on the wetted fraction of the evaporation layer, with the assumption that roots are only in the normally wetted area. The fraction transpiration ($FT_k$) is the fraction of total transpiration that comes from the evaporation layer. This initial water balance results in the calculation of an initial water content, $\theta_{j-1,\ k}$ (Equation (10)), without considering infiltration to the next layer.

$$\theta_{j,k} = \theta_{j-1,k} + (Irr_{surf} + R_{wet} - ET_j\,(FT_k) - E_{wet})/FW_{k/}dz_k \tag{10}$$

where $\theta_{j-1,k}$ is the previous water content in layer k (mL/mL), $\theta_{j,k}$ is the current water content in layer k (mL/mL), $\text{Irr}_{surf}$ is the depth of irrigation water that enters surface layer (m), $ET_j$ is the total transpiration from crop during current time step (m), $FT_k$ is the fraction of transpiration from layer k (m), $E_{wet}$ is the evaporation from the wetted fraction of layer k (m), $R_{wet}$ is the rainfall that reaches the wetted fraction of layer k (m), $FW_k$ is the fraction of layer k wetted in the horizontal direction, and $dz_k$ is the depth of layer k (m).

If $\theta_{j,k}$ exceeds FC, then the soil layer is drained to FC, where $\theta_{j,k} = \theta_{FC,k}$, and excess water is routed to the next layer.

In layers below the upper evaporation layer, Equation (11) is used for the water balance. The term $\text{Irr}_k$ in this case refers to irrigation water allocated directly to layers below the evaporation layer. The term $I_{j,k+1}$ is the infiltration from the layer above. The water flux into the layer is divided by the fraction wetted and layer thickness in order to calculate the change in water content.

$$\theta_{j,k} = \theta_{j-1,k} + (\text{Irr}_k + I_{j,k+1} - ET_j\,FT_k)/FW_{k}/dz_k \tag{11}$$

In this simulation of cotton, the horizontal wetted fraction after the first alternate furrow irrigation was 30% in the upper two layers, 40% in the third layer, and 50% in all other layers. It was 100% in all layers in all other irrigations, which irrigated every furrow.

WINDS input data include weather, irrigation, crop, soil, and remote sensing parameters. All input parameters in the WINDS model were measured in the field experiment except the fraction of transpiration extracted from each soil layer (root activity). The root zone extraction fractions, thus, were calibrated to match the observed data. The extraction fractions in WINDS are specified in a table as a function of root depth.

The weather input data included daily minimum and maximum temperature, relative humidity, rainfall, and $ET_o$. Irrigation input data include irrigation days and depths of application for each plot. Irrigation inputs were carefully measured for each plot in the field experiment, as described earlier in Section 2.1.

The crop input data included lengths of growth stages, day of year (DOY) of planting and harvest, crop height, root depth initial and final values, and $K_{cb}$ values. The $ET_c$ in the WINDS model is calculated with the FAO56 [25] dual-component evapotranspiration model, which separates evaporation and transpiration. The $K_{cb}$ was estimated by NDVI for the NT treatment, based on regression from the previous three readings, and was preset according to an FAO56 crop coefficient curve in the FT treatment. Lengths of FAO56 growth stages and $K_{cb}$ values are provided in Table 3 for the FT treatment; however, the late-season stage was extended for five days to match the actual season length of 160 days. For all plots, the initial and end-of-season $K_{cb}$ values were those provided for cotton in FAO56 [25] (Table 17), which are 0.15 and 0.50, respectively.

The soil profile for each study plot was divided into eight, 20 cm layers, with layer 1 representing the 160–180 cm layer and layer 8 representing the 20–40 cm layer. The top 0 to 10 cm layer is the evaporation layer (layer 9). Soil input data included initial soil water content, FC, WP, and saturated water content for all layers of the soil profile. Mualem–Van Genuchten coefficients were calculated with the Rosetta model based on FC and WP values.

Because soil samples were collected at 30 cm intervals, whereas NMM measurements were at 20 cm intervals, interpolations of laboratory-measured FC and WP values were made to adjust the values to 20 cm. During the simulations of soil water content, about 20% of the measured FC and WP values for soil layers were adjusted based on poor model agreement with observed measurements for certain layers. The FT1 plot needed no adjustment, and for NT1 and NT2, only WP values were adjusted. The adjustments for other plots varied between 2.0 and 6.0%, either higher or lower than the lab-measured value of that layer. After adjustments, the soil water simulation in WINDS was evaluated for accuracy using linear regression of simulated versus measured soil water contents in all soil layers over the growing season and resultant coefficient of determination ($r^2$) and root mean squared error (RMSE) criterion.

*2.4. Sensitivity Analysis*

Sensitivity analysis evaluates the sensitivity of the model to changes in parameters. In this case, we evaluated the effect of changing the Van Genuchten parameters on daily simulated soil water content. These parameters are used in Richards' equation. We also evaluated changes in the tipping bucket model parameters, FC and WP, for two scenarios:

Scenario 1: Model was run, changing values of Mualem–Van Genuchten parameters discretely from sandy loam soil to two extremes of the soil classification; sand and clay (default values), while the tipping bucket parameters were kept the same as that of sandy loam.

Scenario 2: Model was run, changing FC and WP from the values for sandy loam soil to sand and clay (default values) while keeping the Mualem–Van Genuchten parameter values of sandy loam. Table 4 shows the values and the sources for each parameter.

**Table 4.** Values of Mualem–Van Genuchten and tipping bucket parameters for three soil types used in the model validation.

| | **Mualem–Van Genuchten Parameters** | | |
|---|---|---|---|
| | **Sand** [a] | **Sandy Loam** [b] | **Clay** [b] |
| $\alpha$ (m$^{-1}$) | 14.5 | 7.5 | 0.8 |
| N (dimensionless) | 2.6 | 1.89 | 1.09 |
| $\theta$r (%) | 4.5 | 6.5 | 6.8 |
| Ksat (m/day) | 7.128 | 1.061 | 0.038 |
| L (dimensionless) | 0.5 | 0.5 | 0.5 |
| | Tipping bucket parameters | | |
| | Sand [a] | Sandy loam [c] | Clay [c] |
| Field Capacity (%) | 10 | 18.2 | 25.7 |
| Wilting point (%) | 4 | 10.2 | 19.2 |

Notes: [a] values extracted from [43] (Table 8.3); [b] values estimated by the Rosetta model, and [c] values from the field measurements.

## 3. Results

*3.1. Field Experiment Results*

The estimated $K_{cb}$ for the FT treatment was the same for all replicates, while the $K_{cb}$ for NT replicates varied at the end of the season (Table 5). The average initial (0.15) and mid-season (1.20 to 1.22) estimates for $K_{cb}$ were similar for the two treatments. The mid-season $K_{cb}$ using the FAO56 standard 1.15 value adjusted to the local climate was 1.21, or essentially the same as those used in the treatment plot irrigation scheduling. The end-of-season $K_{cb}$ was 0.51 for FT. The estimated end-of-season values for NT plots, based on remote sensing indices, varied from 0.18 to 0.38 with an average of 0.28. The variation in the total estimated seasonal $ET_c$ for the FT replicates (Table 5) was due to a combination of the differences in soil evaporation losses (different TEW values based on soil properties) and reduced $ET_c$ due to soil water stress (different TAW values based on soil properties). This was essentially true for the differences in estimated $ET_c$ of the NT replicates, except with some additional, though slight, variation in estimated $K_{cb}$ due to NDVI variation.

Table 5 also shows the treatment replicate variation of measured maximum crop height, total irrigation applied, measured (actual) $ET_c$, and lint yield (LY) for both $K_{cb}$ methods. The differences in total irrigation applied to treatment replicate occurred primarily during the first week following planting to establish the crop. At that time, irrigation amounts varied in plots according to that needed to satisfactorily "sub" the furrow irrigation water to the seedbed. Average total irrigation amounts between the FT and NT treatments were essentially equal (960 to 967 mm). Similarly, the average measured total $ET_c$ was found to be the same for the NT and FT treatments (988 to 990 mm). The average measured total $ET_c$ was 35 to 48 mm higher than estimated for the NT and FT, respectively. There was a higher variation in total measured $ET_c$ among the replicates of a given treatment. Some differences were related to deep percolation, e.g., the lowest $ET_c$ of the NT was in NT3, which had 40 mm of DP plot compared to 28 mm in NT1.

**Table 5.** Estimated treatment plot Kcb for initial, mid-season, and end-of-season growth stages and estimated total crop evapotranspiration ($ET_{c\,[E]}$) through 1 October (DOY 274); measured maximum crop height, total irrigation (I), $ET_{c\,[M]}$, and Lint Yield (LY) for 2007 cotton experiment. The numbers in bold for the replicates in a treatment represent the highest value of a parameter, and italic represents the lowest value.

| Treatment | $K_{cb}$ | | | $ET_{c\,[E]}$ (mm) | Crop height (m) | I (mm) | $ET_{c\,[m]}$ (mm) | LY (kg/ha) |
|---|---|---|---|---|---|---|---|---|
| | Initial | Mid | End | | | | | |
| FT-1 | 0.150 | 1.20 | 0.51 | 936 | 1.16 | 953 | 988 | 1957 |
| FT-2 | 0.150 | 1.20 | 0.51 | 910 | **1.23** | *928* | *977* | **2016** |
| FT-3 | 0.150 | 1.20 | 0.51 | 955 | 1.19 | **988** | **1012** | 1876 |
| FT-4 | 0.150 | 1.20 | 0.51 | **966** | 1.29 | 972 | 981 | 1876 |
| FT avg | 0.150 | 1.20 | 0.51 | 942 | 1.22 | 960 | 990 | 1931 |
| NT-1 | 0.152 | 1.22 | 0.30 | 961 | 1.19 | 971 | 1007 | 1947 |
| NT-2 | 0.152 | 1.21 | 0.18 | *937* | 1.30 | *954* | 974 | **2040** |
| NT-3 | 0.152 | 1.21 | 0.38 | **964** | *1.06* | **976** | *961* | 1972 |
| NT-4 | 0.152 | 1.21 | 0.27 | 953 | 1.12 | 969 | **1009** | *1851* |
| NT avg | 0.152 | 1.216 | 0.28 | 954 | 1.17 | 967 | 988 | 1953 |

The average treatment LYs were $\approx$ 1940 kg ha$^{-1}$. At the replicate scale, there were no obvious relationships between the final LY and irrigation or water use differences. The total $ET_c$ and LY for the 2007 study compared favorably with other cotton studies in the immediate area for fully irrigated cotton under typical density populations. In several studies [9,27,39,41], total measured $ET_c$ varied from 940 to 1060 mm, and final lint yield varied from 1500 to 2200 kg ha$^{-1}$. The field results for 2007 show that both $K_{cb}$ methods were similar in irrigation, $ET_c$, and yield parameters and are representative of well-managed cotton production in this arid climate.

The general soil type for treatment plots is given in Table 6, which varied from sandy loam to sandy clay loam. A more detailed depiction of soil layer texture for each plot is provided in Figure 3. As for the irrigation and $ET_c$ measures, there was not an obvious correlation between a particular soil type and the LY.

**Table 6.** Predominant soil type for soil layers of the FT and NT treatment plots in 2007 at Maricopa, Arizona.

| Treatment Name | Plot Code | Soil Type |
|---|---|---|
| FT Replicate 1 | FT-1 | Sandy clay loam |
| FT Replicate 2 | FT-2 | Sandy loam |
| FT Replicate 3 | FT-3 | Sandy clay loam |
| FT Replicate 4 | FT-4 | Sandy clay loam |
| NT Replicate 1 | NT-1 | Sandy clay loam |
| NT Replicate 2 | NT-2 | Sandy loam |
| NT Replicate 3 | NT-3 | Sandy clay loam |
| NT Replicate 4 | NT-4 | Sandy loam |

*3.2. Model Results*

The model was calibrated against field experiment soil water content data where the simulated daily water contents were adjusted to bring them in proximity to the NMM reading dates. The primary calibration was for root growth rate and root water extraction fractions during the season; however, soil moisture parameters were adjusted at a few locations and depths out of the eight locations and seven layers per location. Figure 4 shows the WINDS model output for the online version where plot 12 was selected from the drop-down menu (https://vis.datascience.arizona.edu/WINDS/) for soil layer water contents vs. time after planting.

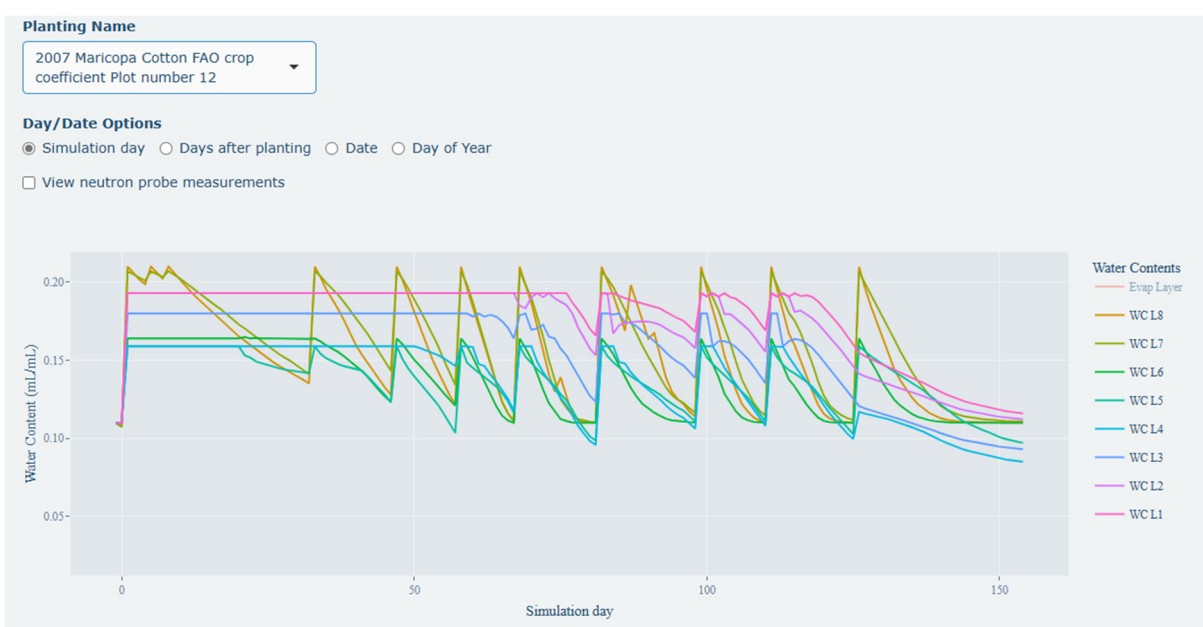

**Figure 4.** WINDS (Water-use, Irrigation, Nitrogen, Drainage, and Salinity) model simulation of soil water content vs. time (https://vis.datascience.arizona.edu/WINDS/ Please wait for web page to load) for soil layers for an FT treatment in the 2007 cotton experiment in Maricopa, Arizona. Soil layers are numbered from the bottom of the soil profile to the top.

The calibrated model showed close agreement with measured soil water contents with the neutron probe (20–180 cm), as shown in Figures 5 and 6 for FT and NT plot replicates, respectively. As expected, the water content in the upper layers showed more temporal variation than in lower layers. This was primarily due to higher ET fractions in the upper layers and the low fraction of water that infiltrates to the lower layers. The close match between simulation and observed value decreases in water content between irrigations demonstrates the value of calibrating the root activity as a function of time after planting.

Figure 7 shows one of the few cases where soil moisture parameters were adjusted due to disagreement between laboratory measurements and observed data. It is the lower layer in the FT4 plot. Figure 7a is the soil water simulation with a laboratory-measured value of FC (23.5%) in the FT4 plot. Reducing the FC to 17.5% resulted in a better fit with the NMM. Adjustment of lower layers such as this has little effect on the model simulation since little water penetrates to the lower layer or was removed by plants.

Table 7 presents the statistical results for WINDS simulation versus measured soil water content data for the FT and NT replicates. The data include all measurement dates and soil layers for the cotton growing season. The $r^2$ for each treatment averaged 0.92, whereas the average RMSE was 0.026 and 0.028 $m^3$ $m^{-3}$ for the FT and NT treatments, respectively (Table 7).

**Table 7.** Root Mean Squared Error (RMSE) and coefficient of determination ($r^2$) for simulated and measured soil water content ($m^3$ $m^{-3}$) for all replicates of FAO and NDVI typical density treatments.

|  | $K_{cb}$ Method | Replicate 1 | Replicate 2 | Replicate 3 | Replicate 4 | Average |
|---|---|---|---|---|---|---|
| RMSE | FAO | 0.022 | 0.026 | 0.026 | 0.032 | 0.026 |
|  | NDVI | 0.025 | 0.032 | 0.029 | 0.029 | 0.028 |
| $R^2$ | FAO | 0.87 | 0.86 | 0.98 | 0.97 | 0.92 |
|  | NDVI | 0.94 | 0.90 | 0.89 | 0.95 | 0.92 |

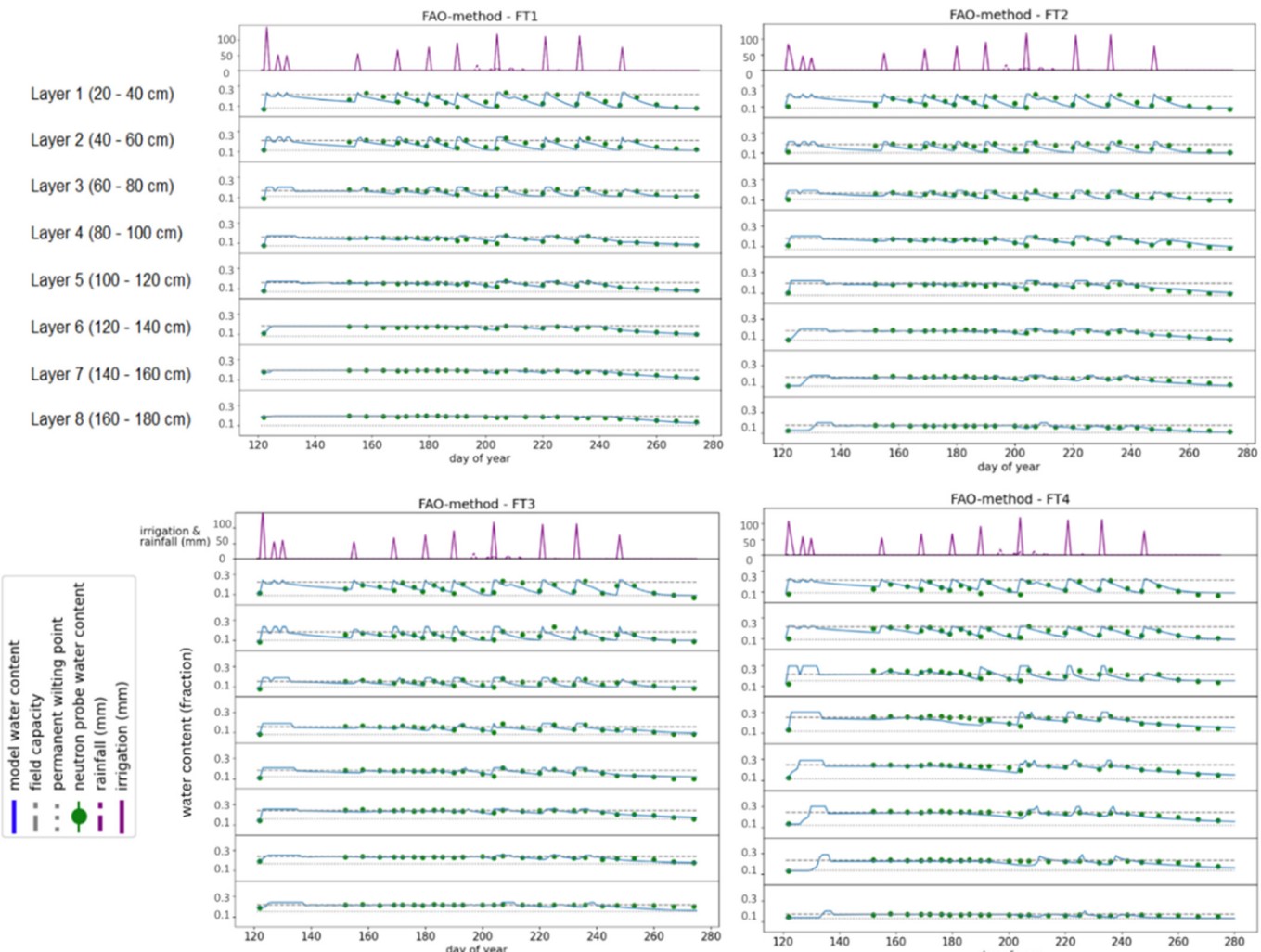

**Figure 5.** Daily simulated water content vs. neutron moisture meter measurements for the FAO (FT) K$_{cb}$ method.

The WINDS model simulates the daily soil evaporation, transpiration, and total ET$_c$ for the growing season (Figure 8). Evaporation was high after irrigations during the initial stage (DOY 122–152) where soil evaporation is the primary component of ET$_c$, while beyond DOY 152, the ET$_c$ is controlled mostly by the transpiration process as the crop develops.

Comparison of the average simulated WINDS ET$_c$ for FT2 with the average ET$_{c\ act}$ (Equation (8)) for periods between two consecutive NMM reading dates during the season showed good agreement (Figure 9). The model ET$_c$ had much less correlation with observed values during the period prior to DOY 200 than from DOY 200 to 275. This was probably caused by the fact that the neutron probe was not sensitive to changes in moisture content in the evaporation layer, and there were periods of high evaporation in the early season. Because the evaporation in the upper layer is a critical component of the overall SWB, improvement in the WINDS ET$_c$ simulation might require TDR probes in the upper soil layer.

A sensitivity analysis for the FT2 treatment showed that the model is sensitive to changes in tipping bucket (FC and WP) parameters (Figures 10 and 11, upper and lower lines). The model was insensitive to changes in Mualem–Van Genuchten's parameters (Figure 11, middle lines) for this sandy loam soil. This is because water redistribution after irrigation in a sandy loam soil takes place within one day, the model time step; thus, Richard's equation has little effect on water movement. Although the Mualem–Van Genuchten parameters were not measured in the laboratory, lack of accuracy in Mualem–Van Genuchten parameters did not significantly change the results.

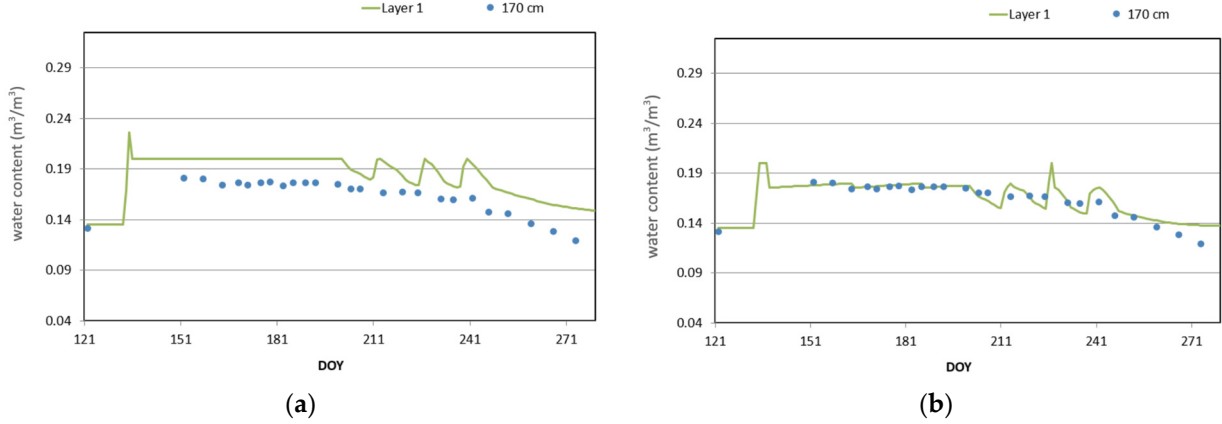

**Figure 6.** Daily simulated water content vs. neutron moisture meter measurements for the NDVI $K_{cb}$ method.

**Figure 7.** Simulation (green) based on laboratory measurement of field capacity (FC) for layer 1 (lower layer) of the FT4 plot (**a**) and simulation with adjusted field capacity (**b**) based on observed neutron probe data (blue circles).

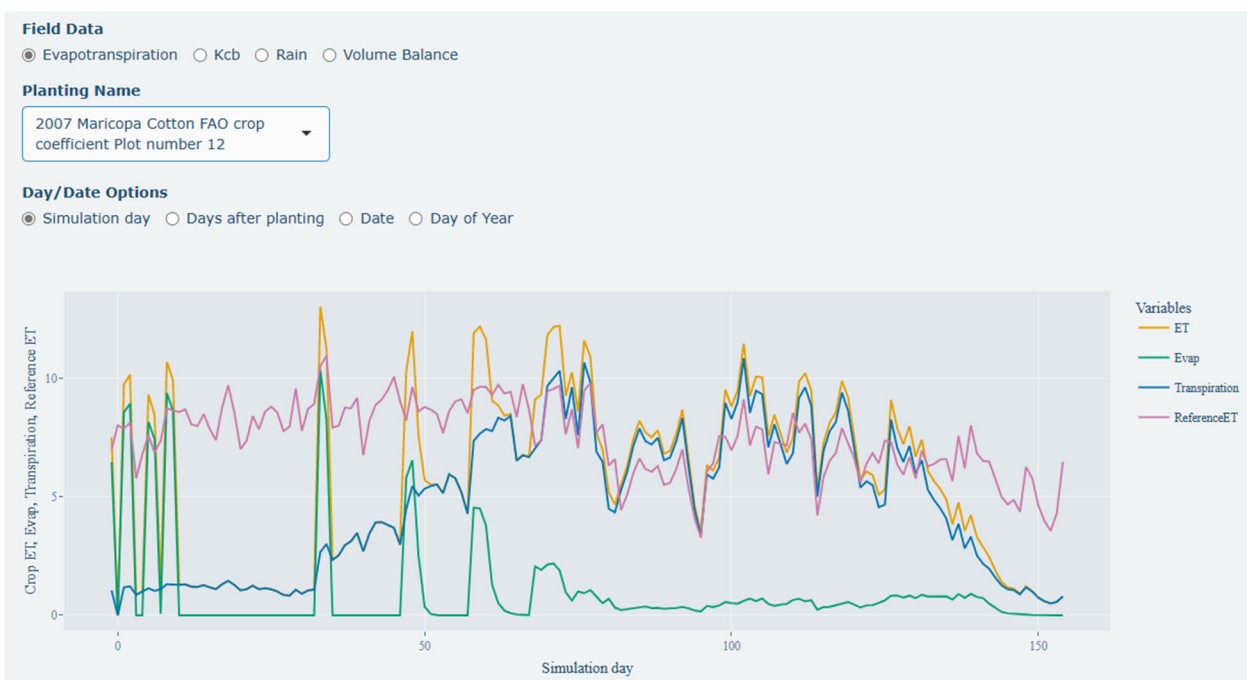

**Figure 8.** WINDS (Water-use, Irrigation, Nitrogen, Drainage, and Salinity) model simulation (https://vis.datascience.arizona.edu/WINDS/ Please wait for web page to load) of crop ET, evaporation, transpiration, and reference ET for the FT (FAO) treatment, plot 12, in the 2007 cotton experiment in Maricopa, Arizona.

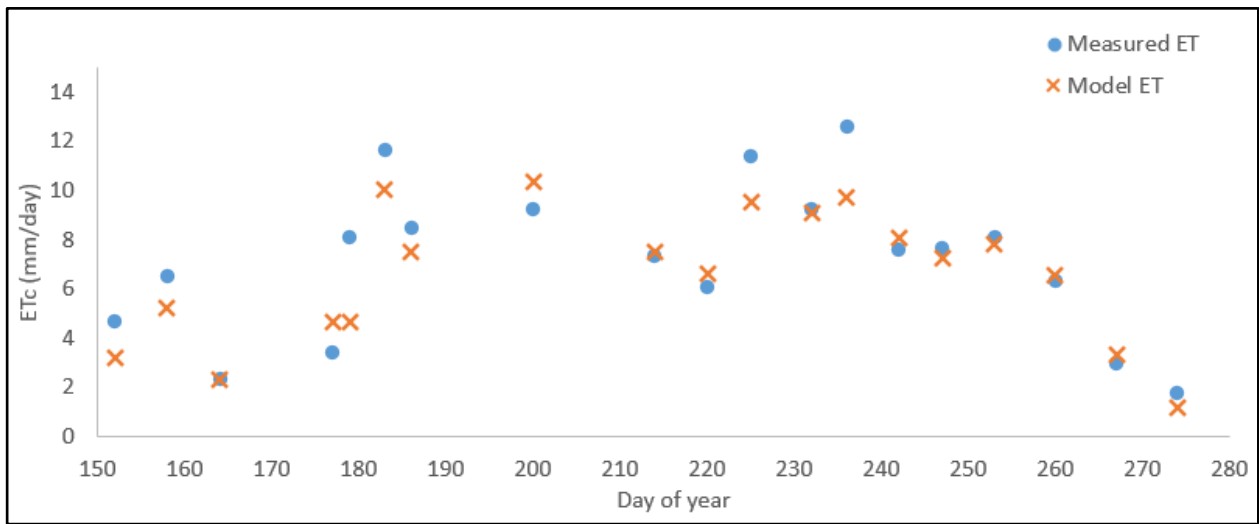

**Figure 9.** Measured ($ET_{c\,act}$) and modeled $ET_c$ comparison between neutron moisture meter measurement intervals for FT2 (FAO crop coefficient curve) treatment.

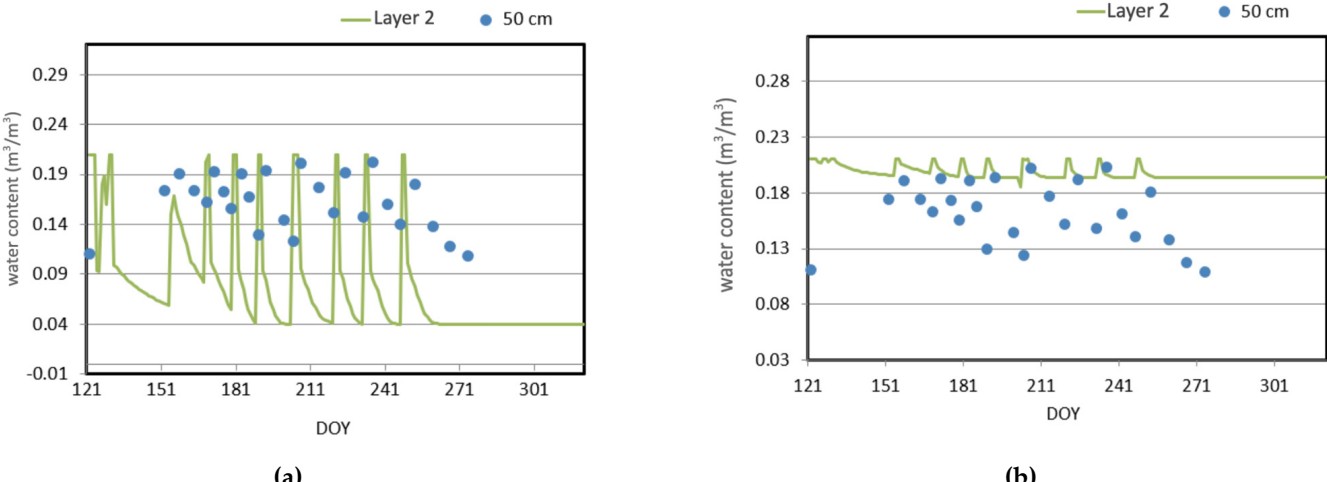

**(a)**

**(b)**

**Figure 10.** Sensitivity analysis of FT2 plot. (**a**) Neutron readings and daily simulated soil water content in layer 2. (**b**) Neutron readings and daily simulated soil water content in layer 2 using tipping bucket parameters for clay.

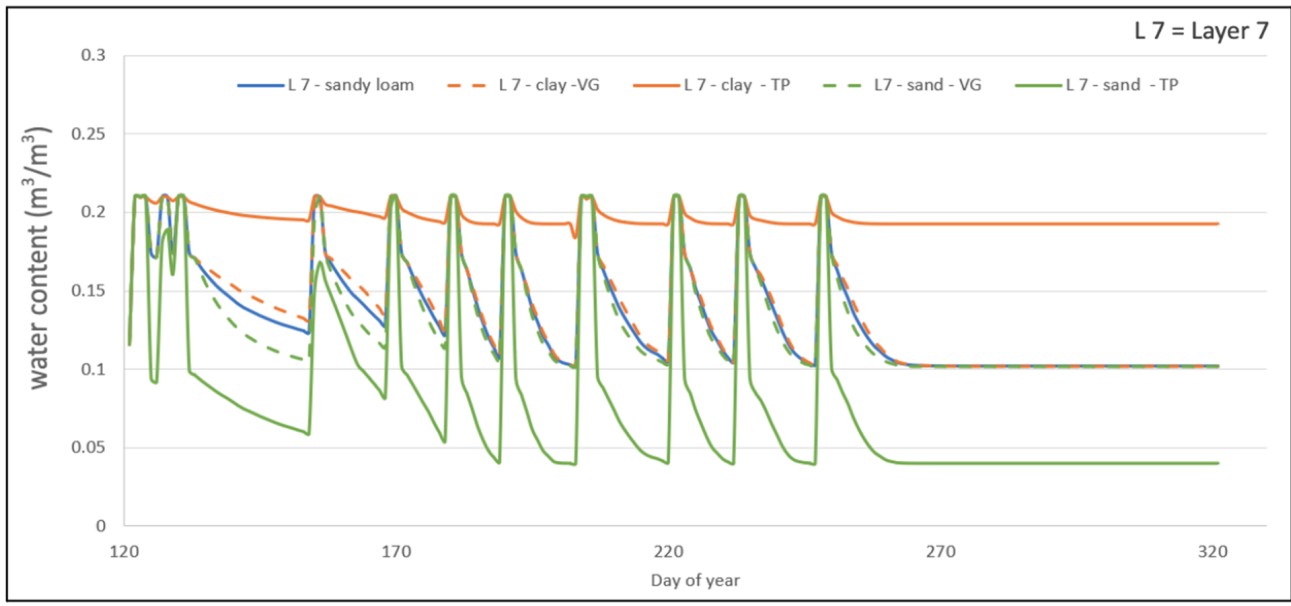

**Figure 11.** Sensitivity analysis for Mualem–Van Genuchten parameters (middle dashed lines), and tipping bucket parameters (FC and WP) (outer solid lines) for FT (FAO crop coefficient curve) treatment.

*3.3. Irrigation Scheduling and Visualization Tool*

The WINDS model has an irrigation scheduling and visualization tool that shows water content as a function of depth in the soil profile and estimates the date of the next irrigation based on recent transpiration from the layers below the upper evaporation layer and current water content in the soil profile (Figure 12). The scroll bar allows the user to scroll to any day during the growing season. The soil profile on the left is color-coded with the range below WP in red, between WP and MAD in yellow, between MAD and FC in green, and between FC and saturation in blue. The irrigation scheduler on the right uses the same colors to represent water content in the entire soil profile.

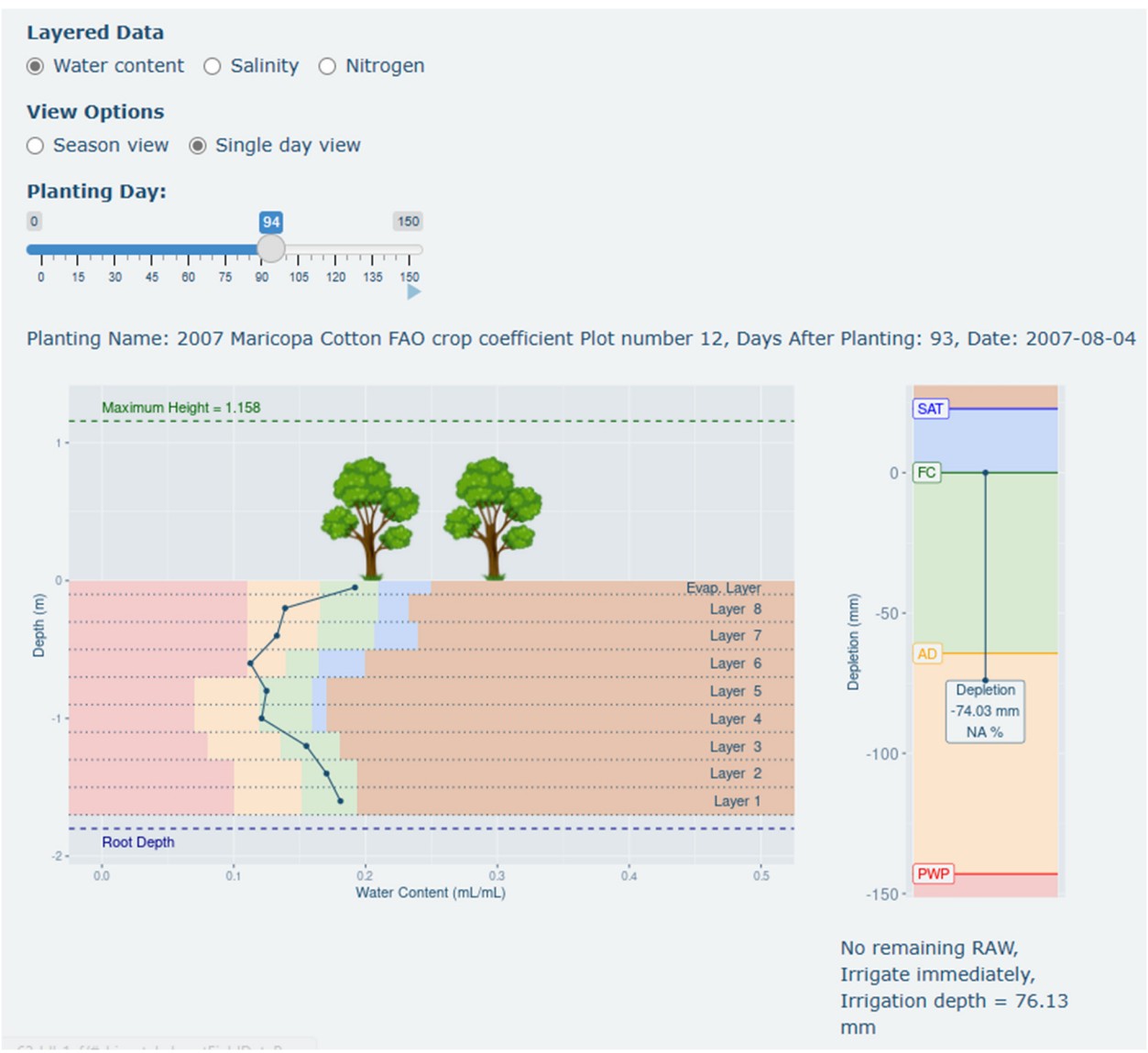

**Figure 12.** WINDS (Water-use, Irrigation, Nitrogen, Drainage, and Salinity) model irrigation scheduling and visualization page (https://vis.datascience.arizona.edu/WINDS/).

## 4. Discussion

The WINDS model is designed to accurately simulate crop water use and uptake in layers, evapotranspiration, irrigation, and soil water content in the arid southwestern United States. This research evaluated the ability of the WINDS model to simulate a cotton crop in sandy loam soils in central Arizona.

The WINDS model had close agreement between observed and simulated evapotranspiration based on weekly neutron probe samples after canopy cover was complete. It is likely that the early season difference between the model and the observed evapotranspiration based on soil water balance was due to the inability of the neutron probe to measure changes in upper boundary soil moisture content due to evaporation.

The WINDS model had close correlation between simulated and observed water content in soil layers. This shows the suitability of the WINDS model for simulation of soil water content in layers for a cotton crop in Arizona soils. It also shows that the WINDS model accurately characterized the root activity (relative water uptake in layers) changes during the growing season. This demonstrates the value and importance of calibrating the root activity as a function of time after planting for specific crops, soils, and irrigation

systems. Future research will determine whether the calibration for one season and soil will apply to other seasons and soils in the area.

The SBAR (Sustainable Bioeconomy for Arid Regions) USDA-NIFA research project has been calibrating the WINDS model to other crops, such as guayule and guar. The advantage of this cotton study is that there were laboratory measurements of soil parameters. At the other sites, FC and WP are determined from observed soil moisture measurements in the field.

We are not aware of other daily time-step models that can provide an accurate representation of water contents within layers throughout the entire soil profile. The ultimate goal is to use WINDS in tandem with soil moisture sensors and remote sensing in order to optimize irrigation, which requires accurate simulation of water content in layers.

To facilitate the use of WINDS in the arid southwestern United States, we added an irrigation scheduling and visualization tool that shows water content as a function of depth in the soil profile and estimates the date of the next irrigation based on recent transpiration from the layers below the upper evaporation layer and current water content in the soil profile (Figure 12). The scroll bar allows the user to scroll to any day during the growing season. The soil profile on the left is color-coded with the range below WP in red, between WP and MAD in yellow, between MAD and FC in green, and between FC and saturation in blue. The irrigation scheduler on the right uses the same colors to represent water content in the entire soil profile.

The innovative aspect of the WINDS model is that it is calibrated for local crops and soils in the arid southwest United States to accurately simulate moisture changes during the growing season across all soil layers due to its root activity model. This study showed that the simulation can be based on laboratory soil moisture parameters rather than just adjusting to observed moisture contents in the field.

## 5. Conclusions

The WINDS (Water, Irrigation, Nitrogen, Drainage, and Salinity) soil water balance (SWB) simulation model can be calibrated to accurately represent soil moisture changes during the growing season. In this study, the WINDS SWB model was calibrated with field data obtained for two basal crop coefficient ($K_{cb}$) treatment methods that were employed in a cotton experiment. One method estimated $K_{cb}$ from a locally derived segmented growth stage curve, and the other estimated $K_{cb}$ from remote sensing data. Experimental results showed that both methods predicted similar total crop evapotranspiration ($ET_c$) and both achieved high cotton lint yields. Thus, the $K_{cb}$ estimates for either method were valid for this region. After calibration, the WINDS model accurately simulated the dynamics of soil water content in sequential soil layers within the soil profile. The only model input data that were changed to improve calibration were the ET fraction table (changes in root activity during the growing season) and a few obviously erroneous soil parameters in a few locations and layers. The sensitivity analysis showed that the WINDS model in sandy loam soil is insensitive to inaccurate estimation of Mualem–Van Genuchten parameters.

**Author Contributions:** Conceptualization, P.W. D.J.H., K.R.T. and A.F.; methodology, P.W., D.J.H. and H.M.; software, P.W. and R.L.; validation, H.M.; formal analysis, P.W., H.M. and D.J.H.; investigation, D.J.H., K.R.T., A.F. and D.E.E.; resources, D.J.H.; data curation, D.E.E.; writing—original draft preparation, H.M.; writing—review and editing, P.W. and D.J.H.; visualization, H.M. and R.L.; supervision, P.W. and D.J.H.; project administration, P.W. and D.J.H.; funding acquisition, D.J.H. and P.W. All authors have read and agreed to the published version of the manuscript.

**Funding:** This project was supported by a Fulbright Degree Program Ph.D. Fellowship for Hadiqa Maqsood, and by the Sustainable Bioeconomy for Arid Regions (SBAR), USDA National Institute of Food and Agriculture (NIFA), USA Grant no. 2017-68005-26867.

**Data Availability Statement:** Data for this research project are available in the Excel spreadsheet WINDS-Cotton-2007.

**Conflicts of Interest:** The authors declare no conflict of interest.

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
