# Peer review of "WINDS Model Demonstration with Field Data from a Furrow-Irrigated Cotton Experiment"

_water, doi:10.3390/w15081544_

Round 1
Reviewer 1 Report
The authors described a cotton irrigation experiment in Arizona, USA. The research concerned the WINDS model and was based on laboratory and field tests.
The authors refer several times to Internet sources, but these links are not available. These references should be corrected or the unavailability explained.
Author Response
The internet links work; however, I added the following statement in two locations. Please wait for web page to load.
Reviewer 2 Report
The whole article has done a very detailed display of the WINDS model, but it also leads to a slightly longer article. In addition, the discussion part and the introduction part of the article do not have a good connection, and further improvements are recommended.
This article has introduced the WINDS model in detail and verified the simulation results.
I highly appreciate the work of the author. Considering that there have been many reports on relevant research directions and that the WINDS model introduced in the article belongs to a very segmented field, I believe that the originality of the article and the interest of readers in the article are limited.The discussion part of the article focuses on the simulation results of the WINDS model, without focusing on the introduction and related research. Therefore, I suggest the author further improve the discussion part of the article. Of course, my suggestions do not affect the publication of the article, as the content of the article already matches the topic very well.
Author Response
Excellent observation about clarification of the purpose of the WINDS model in the introduction.
Although there are excellent daily soil water balance models such as Aquacrop in use around the world, the WINDS model focuses on the irrigation systems, evapotranspiration, soil water status in layers, and crops in the arid southwestern United States.
That was also an excellent observation about tying the Discussion to statements in the Introduction. In the discussion, I made the following statements.
The WINDS model is designed to accurately simulate crop water use and uptake in layers, evapotranspiration, irrigation, and soil water content in the arid southwestern United States. This research evaluated the ability of the WINDS model to simulate a cotton crop in sandy loam soils in central Arizona.
The WINDS model had close agreement between observed and simulated evapotranspiration based on weekly neutron probe samples after canopy cover was complete. It is likely that the early season difference between the model and the observed evapotranspiration based on soil water balance was due to the inability of the neutron probe to measure changes in upper boundary soil moisture content due to evaporation.
The WINDS model had close correlation between simulated and observed water content in soil layers. This shows the suitability of the WINDS model for simulation of soil water content in layers for a cotton crop in Arizona soils. It also shows that the WINDS model accurately characterized the root activity (relative water uptake in layers) changes during the growing season. This demonstrates the value and importance of calibrating the root activity as a function of time after planting for specific crops, soils, and irrigation systems.
I also added the following:
...which requires accurate simulation of water content in layers.
To facilitate the use of WINDS in the arid southwestern United States, we added...